# Transcriptome and Metabolome Analyses Provide Insight into the Glucose-Induced Adipogenesis in Porcine Adipocytes

**Susu Jiang** [1,2], **Guohua Zhang** [2] 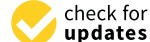, **Jian Miao** [2], **Dianhu Wu** [2], **Ximei Li** [2], **Jiawei Li** [2], **Jianxiong Lu** [2,*] **and Shuangbao Gun** [1,*]

1   College of Animal Science and Technology, Gansu Agricultural University, Lanzhou 730070, China; jiang2019su@163.com
2   College of Life Science and Engineering, Northwest Minzu University, Lanzhou 730030, China; 280112103@xbmu.edu.cn (G.Z.); y210830257@stu.xbmu.edu.cn (J.M.); y210830266@stu.xbmu.edu.cn (D.W.); y220830403@stu.xbmu.edu.cn (X.L.); y220830401@stu.xbmu.edu.cn (J.L.)
*   Correspondence: 165860616@xbmu.edu.cn (J.L.); gunsb@gsau.edu.cn (S.G.)

**Abstract:** Glucose is a major energy substrate for porcine adipocytes and also serves as a regulatory signal for adipogenesis and lipid metabolism. In this study, we combined transcriptome and metabolome analyses to reveal the underlying regulatory mechanisms of high glucose (HG) on adipogenesis by comparing differentially expressed genes (DEGs) and differentially accumulated metabolites (DAMs) identified in porcine adipocytes. Results showed that HG (20 mmol/L) significantly increased fat accumulation in porcine adipocytes compared to low glucose (LG, 5 mmol/L). A total of 843 DEGs and 365 DAMs were identified. Functional enrichment analyses of DEGs found that multiple pathways were related to adipogenesis, lipid metabolism, and immune-inflammatory responses. PPARγ, C/EBPα, ChREBP, and FOS were identified as the key hub genes through module 3 analysis, and PPARγ acted as a central regulator by linking genes involved in lipid metabolism and immune-inflammatory responses. Gene-metabolite networks found that PPARγ-13-HODE was the most important interaction relationship. These results revealed that PPARγ could mediate the cross-talk between adipogenesis and the immune-inflammatory response during adipocyte maturation. This work provides a comprehensive view of the regulatory mechanisms of glucose on adipogenesis in porcine adipocytes.

**Keywords:** pig; adipogenesis; glucose; transcriptome; metabolome; fat accumulation

## 1. Introduction

Adipose tissue is the main metabolic organ for controlling energy and lipid homeostasis and has a wide range of physiological functions. Adipocytes utilize glucose for de novo lipogenesis (DNL), which is critical for mammals that rely on carbohydrates as their primary energy source. Approximately 74% to 77% of body fat in pigs is derived from DNL in the adipocytes using glucose as a substrate [1]. Excessive fat deposition caused by a high carbohydrate diet not only reduces the leanness of pigs, but also affects the flavor and quality of pork [2]. Additionally, excessive energy intake can lead to adipocyte hypertrophy, which is closely related to obesity-related diseases in contemporary society [3]. However, the impact of excess energy intake, particularly glucose, on adipogenesis and adipocyte function remains to be elucidated. Therefore, an in-depth understanding of the regulatory mechanisms of glucose on adipogenesis in porcine adipocytes is vital for controlling body fat deposition and exploring potential targets for obesity treatment.

Adipogenesis is a complex process in which preadipocytes differentiate into mature adipocytes and accumulate fat. This differentiation process is strictly regulated by a series of adipogenic transcriptional factors [4]. These factors, such as peroxisome proliferator-activated receptor gamma (PPARγ), CCAAT/enhancer binding proteins (C/EBPs), and carbohydrate responsive element binding proteins (ChREBP), were considered to be central

regulators of glucose-induced adipogenesis in adipocytes [5,6]. PPARγ is a ligand-activated transcription factor that participates in lipid metabolism, glucose homeostasis, immune responses, and inflammation during different metabolic states [7]. In the adipocytes, PPARγ activation regulates a network of genes involved in free fatty acid uptake and transport, triglyceride synthesis and hydrolysis, β-oxidation, and glycolysis [8]. So, the identification of endogenous PPARγ ligands may provide novel strategies to regulate obesity and metabolic syndrome [9]. Glucose metabolism provides biosynthetic precursors and regulatory signals that drive DNL in adipocytes. ChREBP is a major effector of glucose metabolism, which can be activated by glucose-derived metabolites of glucose-6-phosphate [10]. ChREBP activation induces the expression of genes involved in DNL, glycolysis, and the pentose phosphate pathway [11]. ChREBP—directly or cooperatively with liver X receptor alpha (LXRα) and sterol regulatory element binding protein 1c (SREBP-1c)—promotes glucose-induced adipogenesis in porcine adipocytes [12]. Through control of the generation of endogenous fatty acid species that activate PPARγ, ChREBP links DNL to PPARγ activity and adipocyte differentiation [11,13]. These studies provide valuable knowledge for understanding the mechanism underlying adipogenesis induced by glucose in adipocytes. Therefore, characterizing adipocyte adipogenesis in various glucose environments may offer additional insights into the regulatory mechanisms of adipocyte adipogenesis. In addition, as a response to nutritional overload, adipocyte hypertrophy is closely associated with series of events such as inflammation, lipid metabolism disorders, and insulin resistance [14]. High glucose (HG) accentuates the anti-adipogenic and pro-inflammatory effects of cytokines released by macrophages on human preadipocytes, mediating a crosstalk between adipocytes and macrophages [15]. However, it remains to be studied how immunity and inflammatory responses induced by HG connect with adipogenesis in porcine adipocytes.

Recently, transcriptomics provided a large number of important information for screening potential targets that regulate adipogenesis, which facilitates studies of the regulatory mechanisms of fat deposition and metabolic diseases [16,17]. It was reported that many signaling pathways are involved in glucose and lipid metabolism and cellular energy homeostasis, such as the adenosine-monophosphate-activated protein kinase (AMPK) pathway, cyclic adenosine monophosphate (cAMP) signaling, and wingless-type MMTV integration site (Wnt) signaling [18,19]. Metabolomics could provide reliable information about specific metabolites in adipose tissue from obese patients by systematically studying small molecule metabolite profiles [20]. Although some studies have revealed extensive lipid remodeling in 3T3-L1 or human adipocytes [21,22], as well as the proteome and secretome signatures during adipogenesis [23], further studies are needed to understand the mechanisms underlying adipogenesis induced by glucose in porcine adipocytes. In this study, transcriptomic and metabolomic analyses were conducted to examine the changes in the transcriptional and metabolic profile in porcine adipocytes exposed to HG, in order to screen the differentially expressed genes and differential metabolites, as well as to further explore the relevant metabolic pathways.

## 2. Materials and Methods

### 2.1. Ethics Statement

All animal experiments followed the Regulations for the Administration of Affairs Concerning Experiments Animals (Ministry of Science and Technology, China, 2004), and all procedures were approved and supervised by the Northwest Minzu University Animal Care and Use Committee (Permit No. 0307/2018).

### 2.2. Porcine Preadipocyte Isolation and Cell Culture

Four three-day-old male crossbred piglets (Duroc × Landrace × Large White) from different litters were used (Lanzhou Ruiyuan Agricultural Technology Co., Ltd., Lanzhou, China). Isolation and culture of porcine preadipocytes were performed as described previously [5]. Briefly, the piglets were euthanized through intraperitoneal injection of

sodium pentobarbital at 100 mg/kg body weight. Subcutaneous adipose tissue was isolated from the neck and back of the piglets under sterile conditions. Adipose tissues were rinsed with phosphate-buffered saline (PBS) and chopped into 1 mm$^3$ pieces. Tissue fragments were digested using 0.1% type I collagenase (Solarbio, Beijing, China) in a 37 °C water bath for 1 h. After filtering with a 100-mesh sieve, the mixture was centrifuged for 10 min at 1500 r/min. Following the removal of the supernatant, the sediment was washed with Dulbecco's modified Eagle's Medium/nutrient mixture F-12 (DMEM/F12; Sigma-Aldrich, St Louis, MO, USA) and centrifuged again. The cells were resuspended in DMEM/F12 containing 10% fetal bovine serum (FBS; HyClone, Logan, UT, USA) and seeded in 6-well plates. All cells were maintained in an incubator at 37 °C, 5% $CO_2$, and 95% humidity. After reaching 80% fusion, the cells were digested with 0.25% trypsin (Invitrogen, Carlsbad, CA, USA) and seeded in 12-well plates with a density of $5 \times 10^4$. The medium was changed every two days. When growing to 80% confluence, the cells were cultured in an adipogenic differentiation medium (basal medium supplemented with 100 nmol/L insulin, 1 μmol/L dexamethasone, and 0.5 mmol/L IBMX; Sigma-Aldrich, St. Louis, MO, USA). After adipogenic induction for 2 days, the cells were cultured in a maintenance medium with 5 mmol/L glucose (LG) and 20 mmol/L glucose medium (HG) for an additional 4 days. The cells were collected for gene expression, metabolomics, and RNA-sequencing analyses.

### 2.3. Oil Red O Staining and Quantification of Lipid Content

Adipocytes were stained with Oil Red O as described previously [5]. The cells were washed with ice-cold PBS, then fixed with 4% paraformaldehyde for 30 min. Fixed cells were then incubated in an Oil Red O solution (0.2% Oil Red O in 60% isopropanol) for 30 min. After that, morphologic changes of adipocyts were observed using a microscope (Leica, Wetzlar, Germany). For quantification assessment, the Oil Red O dyes in the cells were extracted into isopropanol and the absorbance was measured at 520 nm.

### 2.4. RNA Sequencing and Transcriptome Data Analysis

Total RNA from cultured cells was isolated using TRIzol regent (Invitrogen, Carlsbad, CA, USA) according to the manufacturer's protocol. The RNA concentration and integrity was confirmed using a Nanodrop 2000 spectrophotometer (Thermo Fisher Scientific, Waltham, MA, USA) and an Agilent 2100 Bioanalyzer (Agilent Technologies, Palo Alto, CA, USA). Samples with an RNA integrity number (RIN) above 8 were used for library preparation. RNA-sequencing libraries were then generated using the NEBNext Ultra II Directional RNA Library Prep Kit (New England Biolabs, Ipswich, MA, USA). Libraries were sequenced on an Illumina NovaSeq 6000 platform (Illumina, San Diego, CA, USA), and 150 bp paired-end reads were generated. Raw data were deposited at NCBI Sequence Read Archive under accession number PRJNA1039275. RNA sequencing analysis was performed by Biomarker Technologies Corporation (Beijing, China) (http://www.biocloud.net, accessed on 10 July 2023).

Raw read quality was evaluated using FastQC software v 0.11.8 [24]. Clean reads were mapped to the porcine genome reference (Sscrofa11.1) by HISAT2 v 2.1.0 [25]. Transcript assembly for each sample was performed with StringTie v 1.3.4 [26]. Gene expression levels were described as fragments per kilobase of exon model per million mapped reads (FPKM). Differentially expressed genes (DEGs) were selected using edgeR v.3.18 with a false discovery rate (FDR) $\leq$ 0.05 and |log2 fold change (FC)| $\geq$ 1.5. Gene Ontology (GO) and Kyoto Encyclopedia of Genes and Genomes (KEGG) enrichment analyses were conducted using Goseq R v 3.15 [27] and the KOBAS package v 3.0 [28], respectively. The GO terms and pathways with *p* values or adjusted *p*-values less than 0.05 were considered significantly enriched.

Expression levels of selected genes were verified by real-time RT-PCR to validate sequencing date accuracy. PCR primers were designed and synthesized by Accurate Biotechnology Co., Ltd. (Hunan, China) (Table S1). The expression of genes was analyzed by the comparative method ($2^{-\Delta\Delta Ct}$) with β-actin as the reference gene.

### 2.5. Construction of Protein–Protein Interactions (PPI) in Network and Module Analysis

The PPI network was constructed by a STRING database v12.0 (https://cn.string-db.org/, accessed on 20 July 2023)with a confidence score of 0.4 (medium level), and then visualized in Cytoscape v 3.7.1. To identify sub-networks and hub genes within the PPI network, the Molecular Complex Detection (MCODE) and CytoHubba plugin in Cytoscape were respectively used [29].

### 2.6. Metabolite Extraction and Metabolome Analysis by Liquid Chromatography–Mass Spectroscopy (LC–MS)

Twelve samples (six samples per treatment) were analyzed using an UPLC-MS/MS system. The samples were dissolved in 1000 μL methanol-acetonitrile solution (1:1 $v/v$), vortexed for 30 s, incubated at $-20\,^{\circ}$C for 1 h, and then centrifuged at $12{,}000\times g$ for 10 min at 4 $^{\circ}$C. A 500 μL supernatant was transferred to a new tube and evaporated with nitrogen. The samples were redissolved in 160 μL of 50% acetonitrile solution, centrifuged for $12{,}000\times g$ at 15 min, and 120 μL of supernatant was loaded into autosampler vials. Additionally, 10 μL of supernatant from each sample was mixed to generate quality control (QC) samples. Metabolome analysis was carried out by Biomarker Technologies Corporation (Beijing, China).

UPLC-MS/MS analyses were performed on an Acquity I-Class UPLS system coupled with Xevo G2-XS QTOF mass spectrometers (Waters, Milford, MA, USA). Chromatographic separation was achieved by an Acquity UPLY HSS T3 column (1.8 μm, 2.1 × 100 mm; Waters, Milford, MA, USA). Data were acquired in both positive and negative ion modes and combined for subsequent analysis. Mobile phase A was 0.1% formic acid solution and mobile phase B was 0.1% formic acid in acetonitrile. The elution gradient program was set as follows: 0.0~0.25 min, 2% B; 0.25~10.0 min, 2~98% B; 10.0~13.0 min, 98% B; 13.1~15.0 min, 2% B. The column temperature was set at 45 $^{\circ}$C, flow rate at 400 μL/min, and the injection volume was 1 μL. Mass spectrometry was performed using a Waters Xevo G2-XS QTOF mass spectrometer with an electrospray ionization (ESI) source operating in positive/negative ion mode. The ESI-MS operating parameters were as follows: source temperature 500 $^{\circ}$C, capillary voltage 2.5 kV (positive) and $-2.5$ kV (negative), gas flow rate 800 L/h, cone gas flow 50 L/h, and cone voltage 30 V.

Raw data were collected using Masslynx v 4.2 software (Waters Corporation, Milford, MA, USA). Progenesis QI v2.3 software (Waters Corporation, Milford, MA, USA) was used for data processing, including peak extraction, alignment, normalization, and identification. Metabolites were identified by searching the Metlin database, Human metabolome database (HMDB), KEGG database, Lipid maps database, and Biomark's self-built databases. Processed data were subjected to multivariate analyses to visualize the distribution of the original data as well as the classification of variables, including principal components analysis (PCA), PLS-Discriminant analysis (PLS-DA), and orthogonal Partial Least Squares-Discriminant Analysis (OPLS-DA) model [30]. The differential accumulated metabolites (DAMs) were evaluated by $|\log_2 \text{FC}| \geq 1$, $p$-value $\leq 0.05$, and the thresholds of variable importance projection (VIP) $\geq 1$. The DAMs were classified according to HMDB and Lipid maps databases. Metabolic pathway analysis of DAMs was annotated by the KEGG database. KEGG pathways with $p$ values < 0.05 were considered significantly enriched.

### 2.7. Combined Metabolomics and RNA Sequencing Analysis

An integrated analysis of metabolomics and transcriptomics data was conducted using the KEGG database, and bubble maps of pathways were generated. KEGG markup language files (KGML, http://www.genome.jp/kegg/xml/, accessed on 15 October 2023) of the pathways were downloaded from the KEGG database, which provides abundant pathway structure information, including nodes (genes and compounds) and edges (functional links) [31]. Based on the KEGG KGML pathway files, relevant aspects of the pathways were curated and combined; The igraph R package was used to create network diagrams showing the relationships between intergroup pathways, genes, and metabolites [32].

*2.8. Statistical Analysis*

The data of real-time RT-PCR and lipid content were presented as the means $\pm$ standard deviation (SD). Quantitative data were visualized with GraphPad Prism 9.0.0 (GraphPad, La Jolla, CA, USA). Differences were considered significant if *p* value < 0.05. Transcriptome and metabolome data analysis was performed using BMKCloud (http://www.biocloud.net, accessed on 9 July 2023).

**3. Results**

*3.1. HG Promoted Adipogenesis in Porcine Adipocytes*

Schematic illustration of the differentiation of adipocytes incubated at different glucose concentrations was shown in Figure 1A. Lipid droplets were visible on day 2, and their abundance gradually increased on day 4 (Figure 1B). By day 4, HG significantly increased adipocyte fat content (Figure 1C). Consistent with previous studies [5], HG significantly promoted the adipogenic differentiation of porcine preadipocytes.

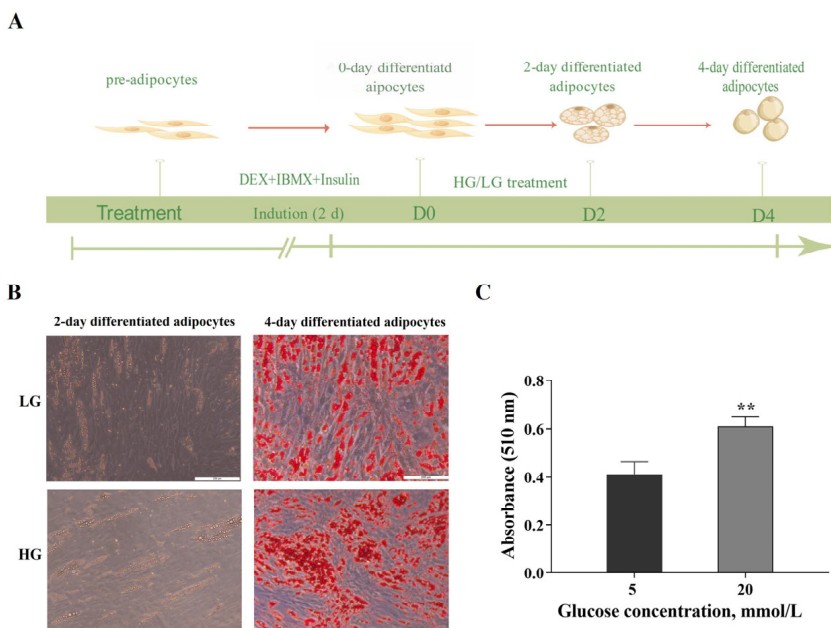

**Figure 1.** Effect of glucose on the adipogenesis in porcine adipocytes. (**A**) Schematic illustration of the differentiation protocol for the adipocytes. (**B**) Differentiating porcine adipocytes exposed to low glucose (LG) or high glucose (HG) for 2 days and for 4 days (Oil Red O staining). (**C**) Quantification of fat accumulation in adipocytes on day 4. ** *p* < 0.01.

*3.2. Identification of DEGs in Porcine Adipocytes*

A total of 200,726,384 clean paired reads from eight libraries were produced after stringent filtration (Table S2). The average Q30 value (Phred quality score > 30) was 95.01% for each library, and the average GC content was 52.80%. More than 95% of the clean reads were mapped to the porcine reference genome, of which 93.10% were uniquely mapped. PCA showed that biological replicates were tightly clustered, and that there was a distinct separation between LG and HG groups (Figure 2A). The heatmap hierarchical clustering indicated a clear differential expression pattern between LG and HG groups (Figure 2C). A total of 843 DEGs were identified in HG compared with LG groups, with 628 upregulated and 215 downregulated genes (Figure 2B, Table S3). To validate the RNA-Seq results, seven randomly selected DEGs were examined by RT-PCR, and the expression patterns of these genes were consistent with the RNA-Seq results (Figure S1), indicating that DEGs identified from the RNA-Seq analysis were reliable.

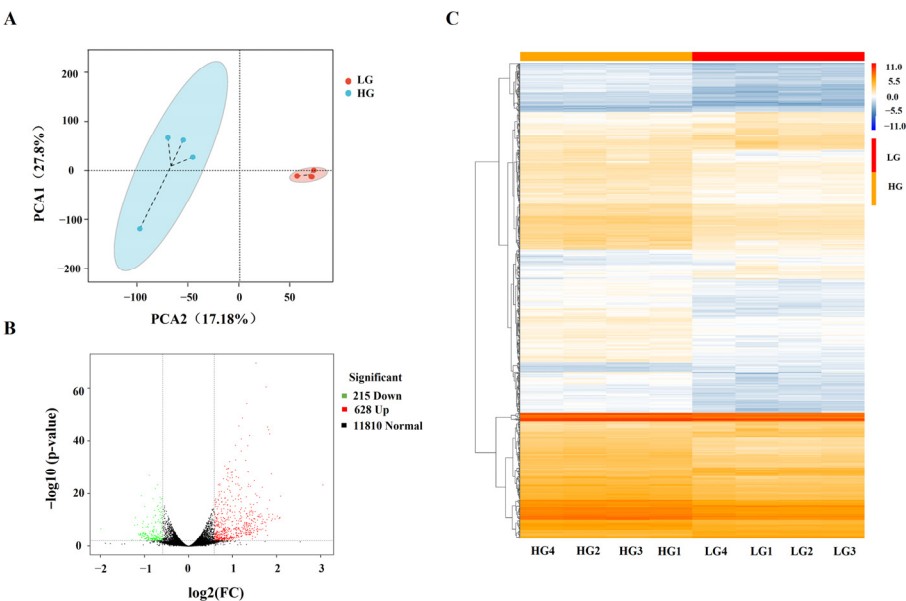

**Figure 2.** Analysis of transcriptome characteristics of adipocytes by RNA sequencing. (**A**) Principal component analysis (PCA) plot of LG and HG groups. (**B**) Heatmap of differentially expressed genes (DEGs) between LG and HG groups. (**C**) Volcanic plot of DEGs between LG and HG groups.

### 3.3. Enrichment and Functional Annotation of DEGs

In the GO functional annotation, the biological processes (BP) category was significantly enriched with 415 GO terms ($p < 0.05$) after HG treatment, followed by molecular function (MF) with 148 GO terms ($p < 0.05$), and cellular component (CC) with 67 GO terms ($p < 0.05$) (Table S4). The cellular process, cell, and binding were the most abundant GO terms within the BP, MF, and CC categories (Figure 3A), respectively. GO enrichment analysis showed that DEGs were primarily enriched in lipid metabolism, carbohydrate metabolism, immune, and inflammation pathways following HG treatment (Figure 3C). GO terms were related to lipid metabolism, namely the very long-chain fatty acid biosynthetic process (GO:0042761), fatty acid biosynthetic process (GO:0006633), fatty acid elongation, saturated fatty acid (GO:0019367), positive regulation of cholesterol efflux (GO:0010875), fatty acid elongase activity (GO:0009922), acetyl-CoA carboxylase activity (GO:0003996), low-density lipoprotein particle binding (GO:0030169), and lipid droplet (GO:0005811).

Based on the KEGG database, the functions of significant pathways were generally consistent with the enriched GO terms (Supplementary Table S5). The top 20 enriched KEGG pathways for DEGs between the LG and HG groups were presented in Figure 3B. Among them, PPAR signaling was the most significantly enriched pathway related to lipid metabolism, followed by regulation of lipolysis in adipocytes, biosynthesis of unsaturated fatty acids, and fatty acid metabolism. In addition, several immune and inflammatory regulatory pathways were identified, including the phagosome, complement, and coagulation cascades, B cell receptor signaling pathway, chemokine signaling pathway, and natural killer cell mediated cytotoxicity.

### 3.4. Identification of the Key Pathways and Hub Genes in the PPI Network

To identify key regulators of complex lipid metabolism pathways, PPI networks were constructed using the STRING database and visualized by Cytoscape. The top three modules included MCODE 1 (MCODE score = 40, consisting of 43 nodes and 1680 edges), MCODE 2 (MCODE score = 15.90, consisting of 20 nodes and 302 edges), and MCODE 3 (MCODE score = 7.24, containing 43 nodes and 304 edges) (Figures 4 and S2). Module one and module two genes were enriched in the cell cycle progression and inflammatory pathways, respectively (Figure S2). The genes in module three were primarily enriched in GO terms involved in immune processes, and in the PPAR signaling pathway via KEGG

pathway annotation. Notably, PPARγ bridged the hub genes involved in inflammation and immune response processes to multiple adipogenic transcription factors and genes, such as PPARγ, CEBPα, FOX, and ChREBP (Figure 4A). The top 10 hub genes identified in PPI network with the highest betweenness centrality were MMP9, TLR2, AGT, PIK3R3, RAC2, PPARγ, PTPRC, FOS, and TYROBP (Figure 4B).

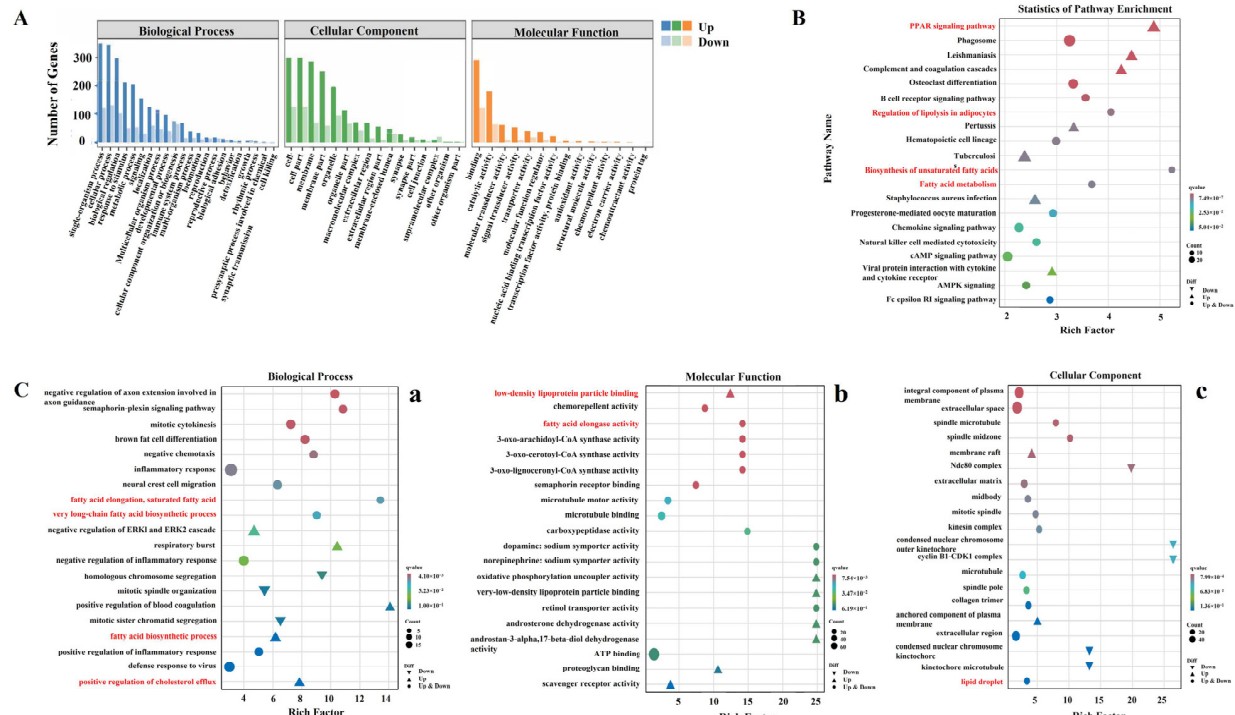

**Figure 3.** GO and KEGG analyses of DEGs. (**A**) GO functional annotation of DEGs. (**B**) Top 20 KEGG pathways of DEGs. (**C**) GO enrichment terms: (**a**) biological process, (**b**) molecular function, and (**c**) cellular components.

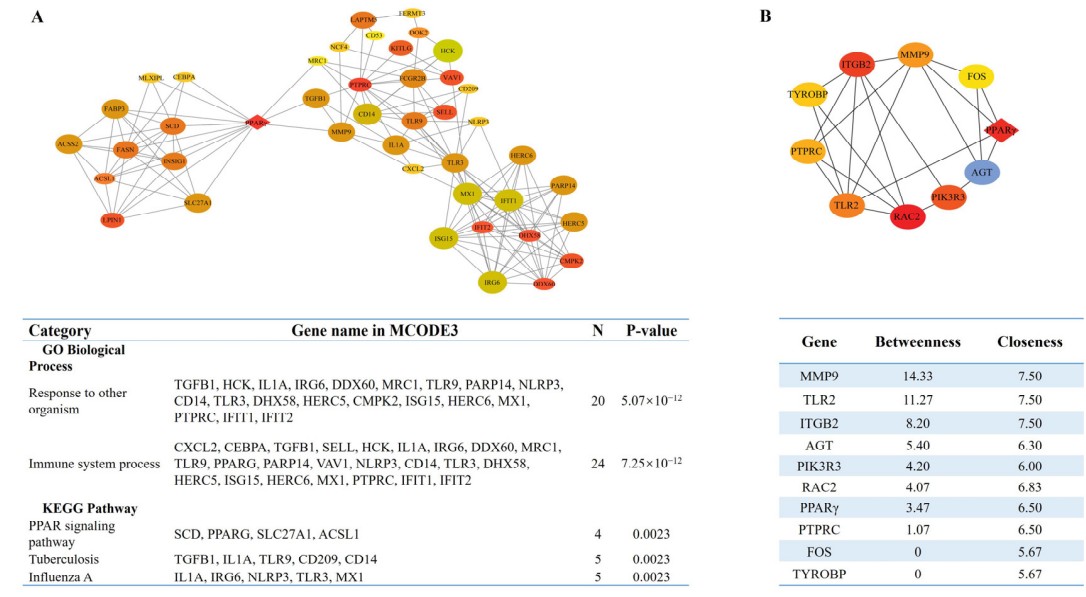

| Category | Gene name in MCODE3 | N | P-value |
|---|---|---|---|
| **GO Biological Process** | | | |
| Response to other organism | TGFB1, HCK, IL1A, IRG6, DDX60, MRC1, TLR9, PARP14, NLRP3, CD14, TLR3, DHX58, HERC5, CMPK2, ISG15, HERC6, MX1, PTPRC, IFIT1, IFIT2 | 20 | 5.07×10⁻¹² |
| Immune system process | CXCL2, CEBPA, TGFB1, SELL, HCK, IL1A, IRG6, DDX60, MRC1, TLR9, PPARG, PARP14, VAV1, NLRP3, CD14, TLR3, DHX58, HERC5, ISG15, HERC6, MX1, PTPRC, IFIT1, IFIT2 | 24 | 7.25×10⁻¹² |
| **KEGG Pathway** | | | |
| PPAR signaling pathway | SCD, PPARG, SLC27A1, ACSL1 | 4 | 0.0023 |
| Tuberculosis | TGFB1, IL1A, TLR9, CD209, CD14 | 5 | 0.0023 |
| Influenza A | IL1A, IRG6, NLRP3, TLR3, MX1 | 5 | 0.0023 |

| Gene | Betweenness | Closeness |
|---|---|---|
| MMP9 | 14.33 | 7.50 |
| TLR2 | 11.27 | 7.50 |
| ITGB2 | 8.20 | 7.50 |
| AGT | 5.40 | 6.30 |
| PIK3R3 | 4.20 | 6.00 |
| RAC2 | 4.07 | 6.83 |
| PPARγ | 3.47 | 6.50 |
| PTPRC | 1.07 | 6.50 |
| FOS | 0 | 5.67 |
| TYROBP | 0 | 5.67 |

**Figure 4.** Identification of Module 3 and the top 10 hub genes in the PPI network. (**A**) Function and pathway enrichment analyses of Modules 3. (**B**) Top 10 hub genes screened from the PPI network. Nodes represent genes and edges represent interactions between DEGs in the PPI network. The colors of the circles (or diamonds) represents different scores (as with Figure 4B).

### 3.5. Identification of Differentially Accumulated Metabolites

Untargeted metabolomics analysis was performed to assay the variation of metabolites in adipocytes exposed to HG. PCA analysis showed that the separation was not complete between LG and HG groups, and the first two principal components (27.55% PC1 and 23.06% PC2) covered 52.61% of the total variance (Figure S3). OPLS-DA, a supervised statistical method, was used to detect metabolite variations in adipocytes (Figure 5A). All samples were inside the 95% confidence interval with obvious separation between the two groups, proven by R 2 Y = 0.995 and Q 2 Y = 0.908. The OPLS-DA model was validated using permutation tests (200 permutations), which confirmed the reliability of the OPLS-DA model (Figure 5B). A total of 365 DAMs were identified, including 233 up-regulated and 132 down-regulated (Figure 5C, Table S6). Heatmap analysis showed that the metabolite profiles were also clearly distinguished between LG and HG groups (Figure 5D).

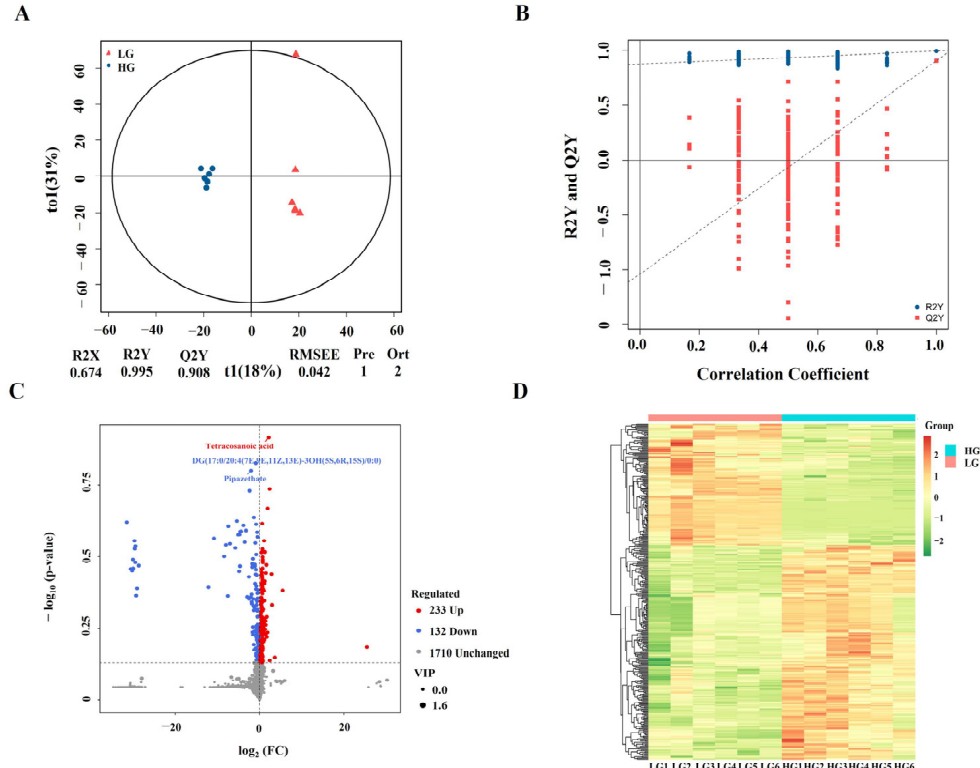

**Figure 5.** Metabolome analysis of the adipocytes. (**A**) OPLS-DA scores plot of metabolic profiles in adipocytes. (**B**) OPLS-DA validation plots. (**C**) Volcano plot of the annotated metabolites. (**D**) Heatmap of differential accumulated metabolites (DAMs).

### 3.6. Pathway Enrichment Analysis of DAMs

Metabolite categories were determined by the HMDB (http://www.hmdb.ca/, accessed on 20 October 2023) and Lipid Maps (https://lipidmaps.org, accessed on 20 October 2023) databases. In the HMDB database, 302 metabolites were classified into 20 categories (Figure 6Aa). Types that were most rich in metabolites were carboxylic acids and derivatives (43 metabolites), glycerophospholipids (41 metabolites), and fatty acyls (31 metabolites). In the Lipid Maps database, 69 metabolites were classified into seven categories (Figure 6Ab). Similar to the results mapped from the HMDB database, the predominant types were fatty acyls (22 metabolites) and glycerophospholipids (37 metabolites). In the fatty acyl family, the significantly increased metabolites included tetracosanoic acid, oleamide, 15-Octadecene-9,11,13-triynoic acid, 9,12,15-Octadecatrien-1-ol, and 13-Hydroxyoctadecadienoic acids (13-HODE), whereas docosapentaenoic acid (22n-3) was decreased (Figure 6Ba). Moreover, the content of glycerophospholipids was signifi-

cantly altered, such as LysoPE [16:1(9Z)/0:0], which was significantly increased; LysoPE [0:0/20:5(5Z,8Z,11Z,14Z,17Z)] was significantly decreased (Figure 6Bb).

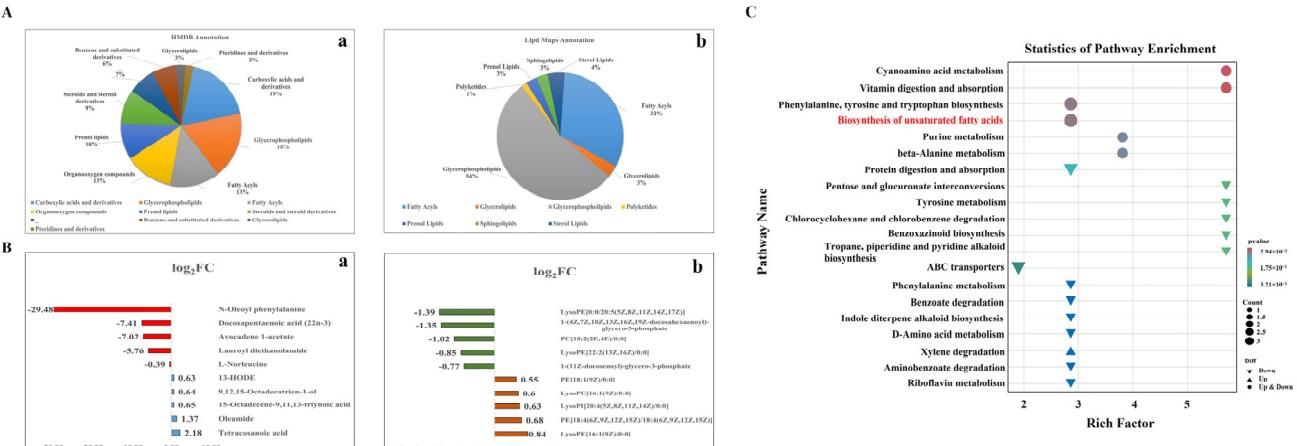

**Figure 6.** Identification of metabolites of adipocytes between LG and HG groups. (**A**) Classification of the DAMs based on HMDB and the Lipid Maps database. (**B**) The significantly altered metabolites in fatty Acyls (**a**) and glycerophospholipids categories (**b**). (**C**) KEGG pathway enrichment analysis of DAMs.

KEGG metabolic pathways for DAMs were assigned to 45 terms between LG and HG groups ($p > 0.05$). These metabolic pathways were primarily related to the metabolism of lipids; amino acid and energy, such as phenylalanine, tyrosine and tryptophan biosynthesis; biosynthesis of unsaturated fatty acids; linoleic acid metabolism; vitamin B6 metabolism; purine metabolism; and beta-Alanine metabolism (Figure 6C, Table S7).

### 3.7. Integrated Analyses of Transcriptomics and Metabolomics Data

To obtain a comprehensive insight into the mechanism underlying adipogenesis and lipid metabolism induced by HG in porcine adipocytes, an integrated analysis of metabolomic and transcriptomic data was performed to explore the related DEGs and DAMs involved in the same pathway (Table S8). A total of 24 integrative KEGG pathways were identified to have both DEGs and DAMs associated with each other (Figure 7A). The results showed that the most DEGs and DAMs were enriched in the PPAR signaling pathway (20 genes and 1 metabolites), biosynthesis of unsaturated fatty acids (7 genes and 3 metabolites), biosynthesis of amino acids (9 genes and 2 metabolites), and purine metabolism (12 genes and 2 metabolites) (Figure 7B), indicating that HG had diverse impacts on both amino acid metabolism and lipid metabolism. Gene–metabolite networks found that 13-HODE may be a key metabolite involved in the PPAR signaling pathway.

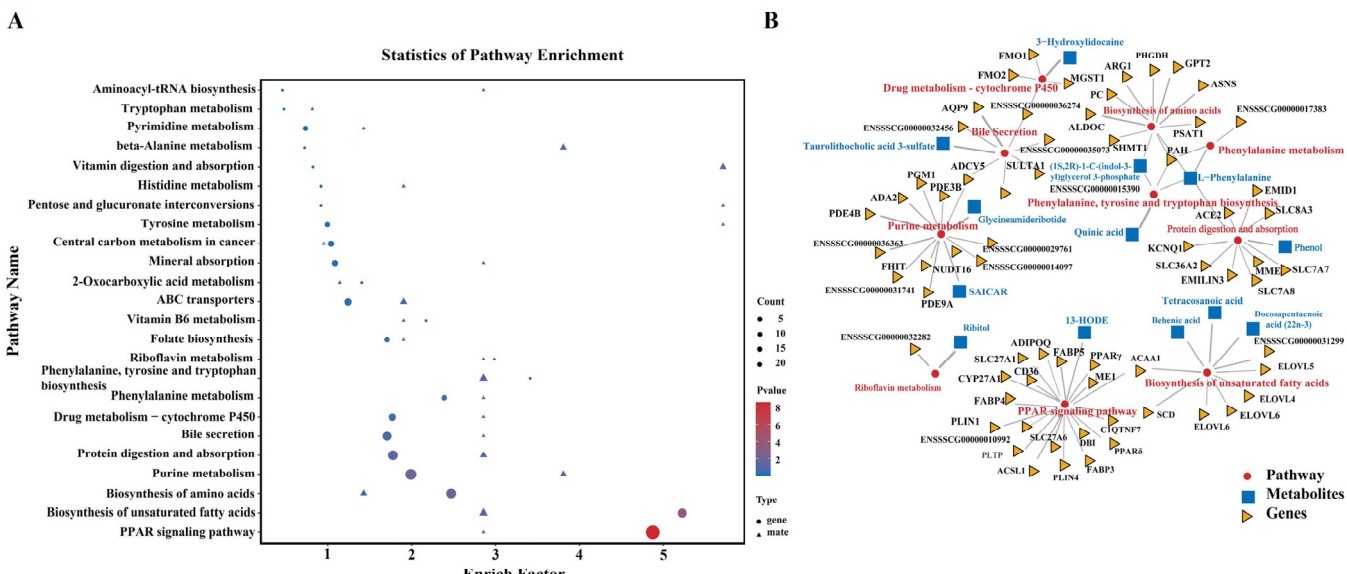

**Figure 7.** Interactive analysis of the metabolome and transcriptome of HG-induced porcine adipocytes. (**A**) Comprehensive analysis of metabolomics and transcriptomics data. (**B**) Network diagram analysis of differential genes and differential metabolites.

## 4. Discussion

Glucose is a main substrate for lipid synthesis in adipocytes, and its metabolic intermediate glucose-6-phosphate enhances the expression and transcriptional activity of ChREBP to promote adipogenesis [5]. In line with previous findings [6], HG significantly increased lipid accumulation in porcine adipocytes. Here, we focused on the regulatory effect of glucose on adipogenesis in porcine adipocytes, and identified 843 DEGs (628 upregulated and 215 downregulated) and 365 DAMs (233 upregulated and 132 downregulated) between LG and HG groups. GO enrichment analysis showed that the DEGs were mainly involved in the inflammatory response, fatty acid elongation, saturated fatty acids, very long-chain fatty acid biosynthetic processes, and the fatty acid biosynthetic process. KEGG pathways analysis showed that PPAR signaling pathways, phagosomes, regulation of lipolysis in adipocytes, complement and coagulation cascades, and biosynthesis of unsaturated fatty acids signaling pathways were significantly enriched. Moreover, integrated metabolomics and transcriptomics analyses found that there were consistent changes in DEGs and DAMs related to lipid and amino acid metabolism in porcine adipocytes exposed to HG, and that the PPARγ-13-HODE interaction network could constitute a complex regulatory network for the regulation of adipogenesis.

The PPAR signaling pathway is a key pathway associated with adipogenic differentiation, fatty acid metabolism, and immuno-inflammatory responses [33]. Ligand-induced activation of PPARs (PPARα, PPARβ/δ, and PPARγ) transmits signals that generate adipogenic effects in adipocytes [7]. In the present study, the upregulated DEGs by HG were significantly enriched in the PPAR signaling pathway, and many of them were direct PPARγ target genes involved in fatty acid synthesis. This supports that the adipogenesis induced by HG in porcine adipocytes depends on PPARγ. PPARδ is a master regulator of fatty acid catabolism in adipose tissue. Activation of PPARδ decreases lipid accumulation in adipose tissue by stimulating expression of genes related to fatty acid oxidation [34]. Therefore, the upregulation of PPARδ by HG seemed to collaborate with PPARγ at the transcriptional level to regulate fat deposition in porcine adipocytes, which could play a role in improving glucose metabolism and lipid homeostasis [35]. In addition, PPARγ was identified as a hub gene and is highly connected to other adipogenic transcription factors, such as C/EBPα, ChREBP, and FOS. This implied that HG-induced adipogenesis could be the outcome of enhanced activity and interaction with multiple transcription factors in porcine adipocytes [5]. C/EBPα, as another potential marker, cooperates with

PPARγ to synergistically trigger the adipocyte terminal differentiation program [36]. It was demonstrated that C/EBPα is essential for adipocyte expansion and the de novo synthesis of fatty acids in response to metabolic challenges [37]. Activated ChREBP mediates the endogenous de novo fatty acid synthesis pathway and affects the transcriptional activity of PPARγ [11,13]. ChREBP-silenced porcine adipocytes still exhibited fat accumulation, suggesting that other transcription factors could be involved in glucose-induced adipogenesis [5]. FOS (known as c-fos gene) is a heterodimer of the AP-1 transcription factor that could contribute to PPARγ activation and adipogenesis, and FOS knockdown resulted in significantly decreased 3T3-L1 adipocyte differentiation [38]. Therefore, HG could regulate adipogenesis in porcine adipocytes, with PPARγ as a core regulator, coordinated with C/EBPα, ChREBP, and FOS. Furthermore, the regulatory networks between the DEGs and DAMs showed that 13-HODE was linked with PPARγ in the PPARγ signaling pathway. These results indicated that 13-HODE could be a key intermediate metabolite regulating lipid metabolism by HG in adipocytes. Previous work established that 13-HODE, an oxidation product of linoleic acid, serves as an endogenous ligand for PPARγ to increase its transcriptional activation [9]. Multiple metabolomics analyses have also indicated that HODE isomers are abundant lipid metabolites in preadipocyte 3T3-L1 cells and in human and mouse plasma [39,40]. Therefore, HG might induce adipogenesis in porcine adipocytes by promoting the generation of metabolite 13-HODE to increase the transcriptional activity of PPARγ. 13-HODE may act as a specific endogenous ligand for PPARγ, and this relationship provides a new understanding of the regulatory mechanism of glucose on adipocyte adipogenesis. Overall, these results suggested that glucose acts as a precursor and a potential signaling molecule to promote the global process of de novo synthesis of fatty acids.

Functional enrichment analysis for the upregulated DEGs showed that the PPAR signaling pathway, biosynthesis of unsaturated fatty acids, and fatty acid metabolism were significantly enriched. These DEGs, such as adenosine triphosphate (ATP)-citrate lyase (ACLY), acetyl-CoA synthetase 2 (ACSS2), acetyl-CoA carboxylase alpha (ACACA, or ACC1), fatty acid synthase (FASN), malic enzyme 1 (ME1), stearoyl-CoA desaturase (SCD), Elongase 2 (ELOVL2), ELOVL5, and ELOVL6, were involved in de novo fatty acid synthesis, fatty acid desaturation, and fatty acid elongation. Acetyl-CoA, the major substrate for DNL, is primarily derived from the conversion of citrates by ACLY [41], as well as from acetate by ACSS2 [42]. In nutrition excess mammals, the glucose-derived pyruvate generates more endogenous acetate [43]. Thus, the upregulation of ACSS2 expression by HG could promote the utilization of glucose-derived carbon to increase the acetyl-CoA pool in porcine adipocytes. ACLY and ACSS2 compensate for each other to a certain extent, but their dominance varies under different nutritional and hypoxic conditions [44,45]. The together upregulation of ACLY and ACSS2 could be an adaptive response of porcine adipocytes to HG [46]. Moreover, ACACA and ME1 are responsible for catalyzing reactions that produce specific non-lipid precursors malonyl-CoA and NADPH, which can subsequently be used by FASN to assemble palmitic acid (C16:0) [47]. Obviously, these steps are critical for the conversion of glucose and its intermediate metabolites into fatty acids. SCD catalyzes the synthesis of monounsaturated fatty acids (MUFA), mainly oleic acid (C18:1) and palmitoleate (C16:1), which was identified as a critical candidate gene for the determination of the fatty acid composition in adipose tissue and muscle tissue [48]. The increases in the synthesis of oleic acid and its derivatives mediated by SCD resulted in the accumulation of more MUFA in pork, as reported in previous studies [49]. ELOVL 6 catalyzes the elongation of saturated fatty acids (SFA) and MUFA with 12 to 18 carbons, and it is considered an important regulator of fatty acid composition, particularly stearic (18:0) and oleic acids (18:1) [50]. Considering the predominance of DNL pathways in porcine adipocytes, upregulation of SCD and Elovl 6 may play a key role in regulating fatty acid composition in response to changes in glucose levels [49].

Moreover, n-3 long chain polyunsaturated fatty acids (n-3 LC-PUFAs), such as docosahexaenoic acid (DHA, 22:6n-3), Eicosapentaenoic acid (EPA, 20:5n-3), and docosapentaenoic

acid (DPA, 22:5n-3), participate in regulating both lipid and carbohydrate metabolism, and must be obtained for mammals from diets. Elovl2 and Elovl5 are involved in n-3 PUFA biosynthesis, especially Elovl2, which elongates DPA to 24:5n-3, a precursor to DHA [51]. KEGG enrichment analysis of DEGs and DAMs showed that the biosynthesis of the unsaturated fatty acids pathway was enriched, which included the upregulated ELOVL2 and ELOVL5, and the downregulated metabolites DPA, suggesting that DHA synthesis was intensified by HG. Interestingly, the DHA level was not affected by HG, while a 7-fold reduction in DPA was observed. Previous studies pointed out that DHA synthesis via ELOVL2 is indispensable for the regulation of DNL and lipid homeostasis in vivo [52], and that lipid and glucose metabolism in mammals was synergistically affected by dietary DHA and sucrose [53]. It might be reasonable to speculate that the upregulation of Elovl2 and Elovl5 by HG was stimulated by insufficient DPA and the demand for DHA in fatty acid synthesis through a feedback mechanism. As a result, HG could stimulate adipogenesis through the upregulation of the whole process of de novo synthesis of fatty acids in porcine adipocytes, and the synthesis of DHA was a prerequisite in the sustained synthesis of fatty acids.

Glycolysis is the primary metabolic pathway for glucose metabolism, and it links glucose metabolism with lipid and amino acid metabolism. Enhancing the glycolytic pathway provides available precursors for fatty acid synthesis. In this study, the carbon metabolism pathway was enriched by DEGs involved in glycolysis, such as glucokinase (GCK), malate dehydrogenase 1 (MDH1), phosphoglycerate dehydrogenase (PHGDH), and phosphoserine aminotransferase 1 (PSAT1) downregulated by HG. GCK phosphorylates glucose to glucose-6-phosphate, which acts as a substrate for DNL [54] and a signaling molecule to activate ChREBP, too. MDH1 and ALDOC participate in the NADH/NAD+ redox balance and aldol condensation reactions during glycolysis, respectively [55,56]. Obviously, increased glycolysis is a key feature of HG-induced fatty acid synthesis in porcine adipocytes. Additionally, the enzymes coded by PHGDH and PSAT1 catalyze de novo serine biosynthesis. The serine synthesis is a side-branch of glycolysis that coordinates anabolic fluxes associated with central carbon metabolism [57]. PHGDH generates glycolytic intermediate 3-phosphoglycerate and controls the flux from the glycolytic pathway into serine synthesis [58]. PSAT1 catalyzes serine synthesis and affects the serine production and its downstream pathways [59]. As a result, HG inhibited serine de novo synthesis by downregulating PHGDH and PSAT1, leading to an increase in the carbon flux flow from the glycolytic pathway to fatty acid synthesis. This notion is supported by recent research which found that inhibition of de novo serine biosynthesis results in hepatic lipid overaccumulation in mice [60]. Furthermore, KEGG analysis showed that downregulated DAMs were enriched in purine, pyrimidine, and amino acid biosynthesis, which may be due to insufficient serine synthesis affecting the biosynthetic pathway using it as a precursor [61]. Collectively, HG could promote DNL through the increase of glycolysis and the inhibition of serine synthesis in porcine adipocytes. It is necessary to further understand the relevance of serine metabolic pathways in adipogenesis induced by HG, which could provide potential avenues for dietary intervention, biomarker discovery, and combating obesity-related diseases [62].

The present study found that the upregulated DEGs were enriched in multiple pathways involved in the immune-inflammatory response, such as the phagosome, complement and coagulation cascades, B cell receptor, and chemokine signalling pathway. These DEGs included more than 20 genes related to antigen processing and presentation, innate immunity and chemokines, which connected immune and inflammatory response with lipid metabolism during adipocyte hypertrophy [23], and were indispensable for the cross-talk between adipocytes and immune cells in adipose tissues [63]. Apparently, HG-induced adipocytes exhibited a pronounced "immune-like" capability that could activate and recruit immune cells to dialogue with themselves [64]. This could be related to the positive role of the immune-inflammatory response in maintaining the adipose tissue microenvironment to promote healthy adipocyte expansion [65]. A recent study also found that

the antigen-presenting functions of adipocytes can improve systemic glucose metabolism in high-fat diet-fed mice [66]. Furthermore, MCODE 3 and hub genes analysis identified that PPARγ was a core linking the hub genes associated with lipid metabolism and immune–inflammatory responses, suggesting that PPARγ could act as a central regulator for the connection between immune–inflammatory responses and adipogenesis during adipocyte maturation.

**5. Conclusions**

In conclusion, the combined transcriptome and metabolome analyses revealed the regulatory mechanisms underlying adipogenesis induced by HG in porcine adipocytes. HG could stimulate adipogenesis in porcine adipocytes through the increase of the whole process of de novo synthesis of fatty acids and glycolysis, as well as the inhibition of serine synthesis, in which the synthesis of DHA was a prerequisite in sustained synthesis of fatty acids. PPARγ, as a core transcription factor, coordinated with C/EBPα, ChREBP, and FOS to mediate HG-induced adipogenesis in adipocytes. PPARγ also was a connecter of the cross-talk between adipogenesis and the immune-inflammatory response, which effected the adipocyte adipogenesis induced by HG. Moreover, 13-HODE was a critical metabolite by HG, serving as an endogenous ligand to enhance transcriptional activity of PPARγ. These findings provide novel perspectives for further exploring the regulatory mechanisms of glucose in lipid metabolism in adipocytes.

**Supplementary Materials:** The following supporting information can be downloaded at: https://www. mdpi.com/article/10.3390/cimb46030131/s1, Figure S1: Validation of seven randomly selected DEGs by RT-qPCR; Figure S2: Function and pathway enrichment analyses of Module 1 (A) and Module 2 (B); Figure S3: Principal component analysis (PCA) of the metabolite profiles for LG and HG groups; Table S1: Primers used for experimtal validation by qRT-PCR analysis; Table S2: Output statistics of sequencing data; Table S3: Summary of DEGs between LG and HG groups; Table S4: Overview of significant enriched GO terms; Table S5: KEGG enrichment analysis of DEGs between LG and HG groups; Table S6: Identification of the differential metabolites between LG and HG groups; Table S7: KEGG anno-tated results of differential metabolites; Table S8: The differential metabolites and genes related top 20 of the enriched KEGG pathways. Figure S1: title; Table S1: title.

**Author Contributions:** Conceptualization, S.J., J.L. (Jianxiong Lu) and G.Z.; methodology, S.J.; software, S.J. and D.W.; validation, S.J., J.L. (Jiawei Li) and G.Z.; formal analysis, S.J.; investigation, S.J.; resources, J.M., X.L. and J.L. (Jiawei Li); data curation, S.J. and J.L. (Jianxiong Lu); writing—original draft preparation, S.J.; writing—review and editing, S.J. and J.L. (Jianxiong Lu); visualization, J.L. (Jianxiong Lu) and G.Z.; supervision, S.G.; project administration, J.L. (Jianxiong lu) and G.Z.; funding acquisition, G.Z. All authors have read and agreed to the published version of the manuscript.

**Funding:** This research was funded by the National Natural Science Foundation of China, grant number 31860633.

**Institutional Review Board Statement:** The animal study protocol was approved by the Northwest Minzu University Animal Care and Use Committee (Permit No. 0307/2018; approval date: 3 July 2018).

**Informed Consent Statement:** Not applicable.

**Data Availability Statement:** The data were submitted to the NCBI Sequence Read Archive (PR-JNA1039275).

**Acknowledgments:** We are grateful to Beijing Biomarker Technologies Corporation (Beijing, China) for assisting in sequencing. All authors have read and agreed to the published version of the manuscript.

**Conflicts of Interest:** The authors declare no conflicts of interest.

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
