# Peer review of "Transcriptome and Metabolome Analyses Provide Insight into the Glucose-Induced Adipogenesis in Porcine Adipocytes"

_cimb, doi:10.3390/cimb46030131_

Round 1

Reviewer 1 Report

Comments and Suggestions for Authors

The article by Jiang et al., shows an exhaustive analysis of the transcriptome and metabolome of porcine adipocytes subjected to a high concentration of glucose. The authors start from the premise that in pigs, as in humans, high glucose content has an important adipogenic effect. The methodology is appropriate and the data analysis methods are proposed that allow establishing results that highlight the main metabolic pathways that are differentially expressed in high glucose medium and, most especially, the stimulation of the expression of proinflammatory molecules, especially those that could activate an immune response associated with obesity.

Only some minor aspects should be considered prior to publishing the work.

1) The authors raise at the beginning of the introduction the potential importance of the model, beyond the knowledge of porcine physiology, due to the similarity in this metabolic aspect between the pig and the human. However, neither in the rest of the introduction nor in the discussion is this aspect treated in depth. I think it is necessary to introduce aspects that allow the results to be interpreted from this comparative perspective or to eliminate this similarity as an important aspect at the beginning of the introduction.

2) In many cases when references are cited for work not carried out on pig adipocytes, the species is not mentioned, given the specific metabolic differences it would be important to do so.

3) There is an excess of abbreviations (including some that are mentioned for the first time in the last paragraphs) that makes reading difficult

4) Line 95 and 387 have some words in red

Reviewer 2 Report

Comments and Suggestions for Authors

This article compared differentially expressed genes and differentially accumulated metebolites found in porcine adipocytes, with an emphasis on combined transcriptome and metabolome analyses, to uncover the underlying regulatory mechanisms of high glucose on adipogenesis. An appealing topic and, all things considered, a superb review. The research plan of the writers was straightforward. There is no confusion in the abstract, and it effectively introduces the various sections of the article. Appropriate references support the content. But it needs to use more up-to-date references; some of them are quite old. The data is accurate, well-organized, and suitable for supporting the article's claims and discussion. The article's stated objectives are adequately addressed in the discussion. It is important to carefully review the references to ensure consistency in the writing and some reference styles need to be corrected.

Reviewer 3 Report

Comments and Suggestions for Authors

The paper provides data about the role of the glucose and particularly high glucose on the adipogenesis in pigs by comparing differentially expressed genes and accumulated metabolites identified in porcine adipocytes. In general, studies about adipogenesis as of great interest partly because this process is closely bonded with health status. However, the authors have to be more convincing about the rationale of the study and to better explain how it is  beyond the state of the art, since many studies on the adipogenesis in pigs in relations to genes exist. Also they should clearly emphasize on the strength and limitations of the methods they applied.  In the present form of the paper, it is difficult for me to judge about the soundness of the experimental design, since the number of animals included in the study is not reported. This is important since the study includes genetic analysis. The sample size/ number of the animals should be mentioned in the Material and method section. The results are clearly presented. The figures and tables are adequate in number. 

The conclusions are supported by the results.

Comments on the Quality of English Language

Minor English language corrections are necessary, mainly concerning the spelling of words.
